# Privacy-Friendly Cross-Domain Recommendation via Distilling User-irrelevant Information

## ABSTRACT

Privacy-preserving Cross-Domain Recommendation (CDR) has been extensively studied to address the cold-start problem using auxiliary source domains while simultaneously protecting sensitive information. However, existing privacy-preserving CDR methods rely heavily on transferring sensitive user embeddings or behaviour logs, which leads to adopt privacy methods to distort the data patterns before transferring it to the target domain. The distorted information can compromise overall performance during the knowledge transfer process. To overcome these challenges, our approach differs from existing privacy-preserving methods that focus on safeguarding user-sensitive information. Instead, we concentrate on distilling transferable knowledge from insensitive item embeddings, which we refer to as **prototypes**. Specifically, we propose a conditional model inversion mechanism to accurately distill prototypes for individual users. We have designed a new data format and corresponding learning paradigm for distilling transferable prototypes from traditional recommendation models using model inversion. These prototypes facilitate bridging the domain shift between distinct source and target domains in a privacy-friendly manner. Additionally, they enable the identification of top-k users in the target domain to substitute for cold-start users prediction. We conduct extensive experiments across large real-world datasets, and the results substantiate the effectiveness of PFCDR. Code is available at https://anonymous.4open.science/r/PFCDR-AE16.

## CCS CONCEPTS

• **Information systems** → **Recommender systems**; • **Computing methodologies** → **Knowledge representation and reasoning**.

## KEYWORDS

Cross-Domain Recommendation; Cold-start Problem; Source-free knowledge distillation

## 1 INTRODUCTION

Cross-domain Recommendation (CDR), aimed at transferring preference knowledge from an auxiliary source domain to the target domain, has shown powerful ability in mitigating the data sparsity challenge inherent in traditional Recommender Systems (RS) [20, 35]. Since CDR scenario has a necessary knowledge transfer process [5], most recent CDR research primarily emphasizes user embeddings and their mapping relationships, relying on a bridge function in the latent space to transfer user preferences [9, 16, 23, 39]. However, sharing original plaintext embeddings or interaction data has become unacceptable due to new privacy regulations such as GDPR [5] being enacted worldwide. Therefore, devising CDR models that safeguard data privacy, including behavior logs and user embeddings, has become an urgent problem.

We focus in this work specifically on realizing the *privacy-preserving* knowledge transfer between the source and target domains. Following previous CDR settings [16, 20, 41], we consider the two domains are partially overlapped in user sets but no intersection in items. Recently, many privacy-preserving CDR [4, 5, 18, 20] adopts the differential privacy mechanism to protect either shared user embeddings or interaction matrices with the target domain. However, it is commonly understood in the field of differential privacy [4, 20] that reducing the privacy budget diminishes the effectiveness of transferred knowledge, while increasing the budget raises privacy risks. Therefore, existing privacy-preserving CDR inevitably face the balance between utility and privacy, which easily incur the suboptimal solutions than traditional CDR. Safeguarding user privacy while maintaining satisfactory performance still remains a substantial challenge in current CDR.

In this work, we identify that the primary cause of privacy leakage in CDR resides in the sharing of original user embeddings or raw user rating information during the knowledge transfer process. This discovery has inspired us to delve into a less-discussed yet intriguing solution: *utilizing only the knowledge embedded in item embeddings instead of users information for CDR to mitigate the risk of compromising user-sensitive privacy while achieving satisfactory performance.* In this context, sharing user-sensitive data, including both user-item interactions and trained user embeddings, is strictly prohibited. Therefore, the privacy-preserving CDR problem has shifted to overcoming the challenge of extracting valuable collaborative filtering signals solely from item embeddings to achieve effective recommendations. For more details about these challenges, please refer to Section 3.2.

To explore this idea, we propose a Privacy-friendly Cross-domain Recommendation model (PFCDR) for privacy-preserving cross-domain recommendation. PFCDR transfers only insensitive prototypes to promote target domain. In this paper, the **prototype** is only distilled from item embeddings, serving as the transferable collaborative filtering signals (For mathematical definition, please refer to Section 4.2). We employ the conditional model inversion mechanism [7, 33] to generate prototypes in a data-free manner. However, implementing model inversion in CDR tasks poses challenges due to the specific nature of the RS model. Firstly, the discrete one-hot representation of RS data hinders gradient calculation. Additionally, a universal learning paradigm is required to accommodate various recommendation models. To address these challenges, we innovatively design a continuous and dense data format to guarantee gradient calculation. Simultaneously, we introduce a new learning paradigm tailored to the new data format to accommodate distinct RS models. Then, we can successfully adopt model inversion on both source and target domains to generate overlapped users' prototypes respectively and capture the nonlinear relationship of prototypes across domains by a bridge function. Finally, for a cold-start user in target domain, we transfer the user prototype from source domain via the pretrained bridge function

and search the top-k users who are well-matched on the transferred prototype for cold-start predicting in the target domain.

The main contributions of our work are summarized as follows.

- We propose a novel Privacy-friendly Cross-domain Recommendation (PFCDR) system that only transfers users' prototypes across different domains without any user-sensitive information.
- We first propose a model inversion mechanism to distil the transferable prototypes by defining a new data format and a new learning paradigm, which can be optimized in the source model instead of using the one-hot RS data. Then, the users' prototypes of the overlapping users are leveraged to generate the bridge function to capture the relationship across domains. Finally, we search top-k users who are well-matched with the transferred prototype for further prediction.
- We establish a theoretical analysis proving that our proposed PFCDR can efficiently protect user privacy. Further, we conducted extensive experiments over three popular cross-domain tasks to evaluate the effectiveness and robustness of PFCDR in cold-start settings. The results consistently show that PFCDR outperforms existing CDR/privacy-preserving CDR baselines.

## 2 RELATED WORKS

### 2.1 Cross-domain Recommendation

**CDR without Privacy Concern.** At the very beginning, CMF [27] assumes the overlapping users' embedding matrix is shared among domains, and the training of overlapping users adopts the data from all domains. Then, researchers proposed neural network-based models to further enhance knowledge transfer ability [11, 15, 30, 32, 42]. To be specific, DDTCDR [17] proposes a latent orthogonal mapping to extract user preferences across several domains while retaining user relationships across distinct latent spaces. In recent years, several CDR methods have been proposed to model different correlations between source and target domains via bridge functions [16, 23, 26, 36, 37, 40, 41]. EMCDR [23] and SSCDR [16] proposed a general CDR framework to learn a user bridge function between the source and target domain. PTUPCDR [41] involved a meta network to learn the personalized bridge function by extra user interactions in the source domain as prior, which gains significant improvement.

**CDR with Privacy Concern.** However, the conventional bridge-based approaches ignore the impracticality of direct access to user-sensitive information in the source domain. Therefore, some researchers focus on privacy-preserving CDR problem. Recent privacy-preserving CDR predominantly fall into two categories, i.e., federated-based [19, 24] and differential private-based[4, 5, 20]. The former employs a federated learning framework to safeguard user information at the client side and constructs a cross-domain embedding transformation model at the server side. The latter adopts a differential privacy mechanism to securely share either user embeddings [20] or interaction matrices [4] with the target domain. These methods inevitably involve transferring sensitive user information perturbed by noise, leading to a dilemma between utility and privacy, which may incur the suboptimal solutions than traditional CDR. The work in [10] shares a similar idea with ours by utilizing item embeddings for CDR. However, it assumes that the two domains have overlapping item sets, which is inconsistent with the most common CDR setting, where there are no overlapping items

between the distinct domains. The paper [18] employs generative adversarial networks (GANs) to generate synthetic rating matrices as a means of protecting the original rating matrix. However, GANs are notoriously difficult to train and are prone to mode collapse.

In summary, the traditional CDR cannot guarantee the privacy of user sensitive information, while privacy-preserving CDR struggles to achieve satisfactory performance. In this paper, we introduce a privacy-friendly knowledge distillation framework to meet the privacy requirements while achieving SoTA performance.

### 2.2 Data-free Knowledge Distillation

Knowledge distillation (KD) is the technique to compress knowledge from one or multi-teacher models into an initialized student [1, 14]. The data-driven KD methods are challenging to practice if the training data is not accessible. Then, to overcome the barrier of large datasets or privacy concerns, data-free KD [7, 21, 43] has been proposed to deal with knowledge transfer via pseudo-data synthesis to train students without using any real data. The interest in data-free KD has grown exponentially in the CV [28] and NLP [22] research community due to their more practical setting and satisfying performance. However, a topic of high practical value but has yet to be explored in the recommendation community.

**Model Inversion.** Model inversion was first proposed by [8], aiming to steal recognizable images from models, especially when attackers lack training data information. Subsequent works found that it can recover training data information starting from random noise with the additional image prior [25] and statistical regularization [33], which highly fulfil the data-free setting. Nowadays, model inversion has become a vital strategy for data-free knowledge distillation. It aims to 'invert' a pretrained model to recover training data $x'$ starting from random noise as an alternative to inaccessible original data $x$ [33].

To our knowledge, this paper is the first to utilize the model inversion paradigm for Cross-Domain Recommendation (CDR) without transferring any user-sensitive information.

## 3 PRELIMINARIES AND CHALLENGES

### 3.1 Problem Setting

Considering a general CDR scenario that two domains have a partially shared user set, but there is no item intersection. Each domain has a user set $U = \{u_1, u_2, ...\}$ and item set $V = \{v_1, v_2, ...\}$ and a rating matrix $\mathbf{R}^{|U| \times |V|}$. $r_{ij} \in \mathbf{R}$ denotes the interaction of user $u_i$ to the item $v_j$. To distinguish the source and target domains, we denote the user and item sets of the source domain as $U^s$ and $V^s$, with the rating matrix represented as $\mathbf{R}^s$. Similarly, for the target domain, we have $U^t$, $V^t$, and $\mathbf{R}^t$. Finally, the set of overlapping users between two domains by $U^o = U^s \cap U^t$.

Suppose we have a one-hot interaction instance $x = (\mathbf{u}_i, \mathbf{v}_j)$, where $\mathbf{u}_i \in \{0, 1\}^{|U|}$ and $\mathbf{v}_j \in \{0, 1\}^{|V|}$. In this representation, only the element corresponding to that index is 1 and all others are 0. Then, in latent factor models, the one-hot representation of an input instance is mapped into low-dimensional dense vectors through embeddings techniques as follows:

$$(\mathbf{u}_i, \mathbf{v}_j)\mathbf{E} = (\mathbf{U}_i, \mathbf{V}_j), \tag{1}$$

where $\mathbf{E} = \{\mathbf{U}, \mathbf{V}\}$ is the embedding matrix, $\mathbf{U}^{d \times |U|}$ denotes the user embedding matrix where the $i$-th column $\mathbf{U}_i$ represents the embedding vector of user $u_i$, and $\mathbf{V}^{d \times |V|}$ denotes the item embedding matrix where the $j$-th column $\mathbf{V}_j$ represents the embedding vector of item $v_j$. $d$ is the dimension size of embedding.

## 3.2 Challenges

In this paper, our objective is to extract informative knowledge for CDR without transmitting any user-sensitive data to the target domain. Therefore, it is natural to leverage the item embeddings, which encapsulate rich collaborative filtering signals within their parameters. However, solely relying on item embeddings for privacy-preserving CDR presents several challenges:

**CH1: How can we extract transferable knowledge from the item embedding matrix to improve recommendations for cold-start users, particularly in scenarios where there is no overlap in item sets between the two domains?** The collaborative training mechanism of RS encodes latent item knowledge and collaborative filtering knowledge in item embeddings [29]. Hence, extracting personalized and transferable knowledge from information-coupled item embeddings for each cold-start user poses a significant challenge.

**Our Solution:** We propose a simple but efficient conditional model inversion mechanism to directly distill the transferable knowledge, while we unify the format of the transferable knowledge and name it as **prototype**.

**CH2: How to address the domain shift between source and target domains without the help of overlapped user embedding or behavior logs?** In CDR scenarios, overlapping users are often used to bridge the domain shift. At the same time, the source and target domains typically cater to different tasks and lack overlapping item sets. Therefore, effectively bridging the domain shift solely based on prototypes presents considerable challenges.

**Our Solution:** We simultaneously distill overlapping users prototypes in the source and target domains and introduce a prototype bridge function to address the domain shift problem.

**CH3. How to predict the preferences of cold-start users in the target domain using transferred prototypes from the source domain?** Existing methods transfer user embeddings from the source domain to predict the rating preferences of cold-start users in the target domain. It poses a challenge to adopt prototype, the item side knowledge distilled from the item embedding matrix, to predict the cold-start user's preference.

**Our Solution:** Since one user typically has similar preferences across different domains, we adopt the transferred prototype from the source domain to identify the top-k similar users in the target domain, thereby predicting the preferences of cold-start users.

## 4 METHOD

The workflow (Figure 1) of the proposed PFCDR is the following: We first rely on the model inversion mechanism to distil the users' prototypes in both the source and target domains (Section 4.2). Then, a bridge function is trained with users' prototypes to align the gap across domains (Section 4.3). Finally, we adopt the trained bridge function to transfer the user prototype from source to target,

and then we search the top-k matched users in target domain for further prediction (Section 4.4).

## 4.1 New Data Format and Learning Paradigm for Adopting Model Inversion

Model inversion typically involves class-conditional generation to extract data information from a pretrained model. For example, given a randomly initialized input $x'$ and a randomly assigned target label $r$, the random input $x'$ is optimized by:

$$(x^*|r) = \min_{x'} \mathcal{L}(f_\theta(x'), r), \quad (2)$$

where $f_\theta$ denotes the pretrained model parameterized by $\theta$, $\mathcal{L}$ is predefined error function and $x^*$ is the optimal solution.

In a recommendation scenario, model inversion seeks to minimize the loss function as follows:

$$(\mathbf{u}_i^*, \mathbf{v}_j^*|r) = \min_{(\mathbf{u}_i, \mathbf{v}_j)} \mathcal{L}(f_\theta(\mathbf{u}_i, \mathbf{v}_j), r), \quad (3)$$

where an interaction instance $(\mathbf{u}_i, \mathbf{v}_j)$ is optimized to match the randomly assigned preference rating $r$ under the constraints imposed by the pretrained recommendation model $f_\theta$.

**Discussion**: Conventional recommendation models typically adopt a lookup embedding table to return embeddings based on indices. However, this process prevents the input data from receiving gradients for optimization. To enable the calculated gradients to propagate back to the input, we have designed a new data format and corresponding learning paradigm for distilling transferable knowledge using model inversion.

*4.1.1 The new data format.* The new data format should fulfil the following constraints: (1) It should be in a differentiable manner to be successfully optimized during inversion. (2) Considering the generality of the proposed PFCDR, the new data format should adapt to different recommendation models. Inspired by the embedding-based recommendation model, which adopts an embedding matrix to map the one-hot representation of inputs into low-dimensional embeddings. We define the dimension of each new data format as identical to the one-hot representation of the original real data. The new data format can be recognized as:

$$\mathbf{a} = (w_1^u, w_2^u, ..., w_{|U|}^u), \mathbf{b} = (w_1^v, w_2^v, ..., w_{|V|}^v). \quad (4)$$

Unlike the one-hot representation of the original data, where only the element corresponding to the index is 1 and all others are 0, the new data format assigns a random weight to each element. The initialization of this new data format for user and item fields is defined by the following formula:

$$w_i^u = \frac{exp(w_i')}{\sum_{i=1}^{|U|} exp(w_i')}, w_j^v = \frac{exp(w_j')}{\sum_{j=1}^{|V|} exp(w_j')}, \text{where } w' \sim \mathcal{U}(0, 1). \quad (5)$$

Each $w'$ is sampled from the Normal distribution for initialization. Then, we adopt softmax to normalize the random sampled $w'$ as the initial form of our new data format to be optimized during inversion. The softmax normalization ensures that the initialized new data format is on the same scale as the original recommendation data, which helps stabilize the optimization process. With our

**Figure 1: PFCDR utilizes model inversion on both the source and target domains to distil users' prototypes for CDR without transferring any user-sensitive data. Then, we employ a bridge function to overcome the domain shift problem across domains. Finally, when we provide a recommendation for a cold user in the target domain, we transfer the prototype $\mathcal{P}_i^s$ to target domain and search the top-k users who are well-matched with the transferred prototype $\hat{\mathcal{P}}_i^t$ for the prediction propose.**

new data format, implementing model inversion on the pretrained RS model proceeds as follows:

$$(\mathbf{a}^*, \mathbf{b}^* | r) = \min_{(\mathbf{a}, \mathbf{b})} \mathcal{L}(f_\theta(\mathbf{a}, \mathbf{b}), r), \tag{6}$$

where $(\mathbf{a}^*, \mathbf{b}^*)$ is the optimal solution on the pretrained RS.

*4.1.2 The additional learning paradigm for adopting new data format.* The existing learning paradigm of the RS model does not support the new data format. Consider MF, which is the cornerstone of embedding-based RS models, as an example. When original RS data $(\mathbf{u}_i, \mathbf{v}_j, r_{ij})$ enters the forward process, MF will transform it into the representation $(u_i, v_j)\mathbf{E} = (\mathbf{U}_i, \mathbf{V}_j)$. With this mapped representation, MF can be trained using the following:

$$\min_{\mathbf{E}} \sum_{(u_i, v_j) \in D} \frac{1}{2|D|} (r_{ij} - \mathbf{U}_i^\top \mathbf{V}_j)^2. \tag{7}$$

Where $D$ is the dataset. $|D|$ denotes the number of training samples. When the input data is our new data format $(\mathbf{a}, \mathbf{b})$, the feed-forward procedure will assign weight to all item embeddings, and the intermediate representation is the following form:

$$(\mathbf{a}, \mathbf{b})\mathbf{E} = (w_1^u \mathbf{U}_1, w_2^u \mathbf{U}_2, ...w_{|U|}^u \mathbf{U}_{|U|}, w_1^v \mathbf{V}_1, w_2^v \mathbf{V}_2, ...w_{|V|}^v \mathbf{V}_{|V|}). \tag{8}$$

For user field, we obtain $(w_1^u \mathbf{U}_1, w_2^u \mathbf{U}_2, ...w_{|U|}^u \mathbf{U}_{|U|})$, a $d \times |U|$ embedding matrix. For item field, we obtain $(w_1^v \mathbf{V}_1, w_2^v \mathbf{V}_2, ...w_{|V|}^v \mathbf{V}_{|V|})$, a $d \times |V|$ embedding matrix. The obtained embedding matrix from data $(\mathbf{a}, \mathbf{b})$ cannot be directly used in the subsequent feed-forward process due to dimensional inconsistency. To enable end-to-end optimization with new data format during inversion, we add an

additional computation module that compresses the entire user embedding matrix $\mathbf{U}$ and item embedding matrix $\mathbf{V}$ through a weighted sum using the data vectors $\mathbf{a}$ and $\mathbf{b}$, respectively. The final learning paradigm of model inversion on pretrained MF is defined as follows:

$$(\mathbf{a}^*, \mathbf{b}^* | r) = \min_{(\mathbf{a}^*, \mathbf{b}^*)} \frac{1}{2} (r - (\sum_{i=1}^{|U|} w_i^u \mathbf{U}_i)^\top (\sum_{j=1}^{|V|} w_j^v \mathbf{V}_j))^2). \tag{9}$$

Like in MF, other embedding-based RS models can optimize the new data format using the same operation without altering the original model architecture.

## 4.2 The Generation of Users' Prototypes

In this section, we define the term **prototype** in the context of our PFCDR and provide a detailed explanation of how users' prototypes are generated using conditional model inversion.

*Definition 4.1 (User Prototype).* Let $\mathbf{b}_i^* \in \mathbb{R}^{1 \times |V|}$ be the optimized weight vector distilled from pretrained item embedding matrix through model inversion conditioned on user $u_i$. The user $u_i$'s prototype is the compression of entire item embeddings $\mathbf{V} \in \mathbb{R}^{d \times |V|}$ through a weighted sum by the distilled $\mathbf{b}_i^*$, i.e., $\mathcal{P}_i = \sum_{j=1}^{|V|} w_{j,i}^* \mathbf{V}_j$.

Equation 6 shows an potential solution to generate users' prototypes by distilling informative knowledge from a pretrained model. However, with only a predefined rating $r$ as a constraint, the distilled information is unpredictable and may mix different users' or items' knowledge. Hence, providing a more instructive strategy

 

to distil the transferable characteristics of each user is essential to further construct users' prototypes.

Intuitively, a user's rating preference is suitable to represent the main characteristics of that user. For example, in [41], the rating preference of user $u_i$ is the weight sum of item embeddings that rated by this user. In this study, we utilize a pretrained model conditioned on a particular user and a chosen target rating to extract user-related knowledge via model inversion. This process involves freezing the gradient of $\mathbf{u}_i$, and the chosen target rating $r$, while solely optimizing the randomly initialized vector $\mathbf{b}$:

$$(\mathbf{b}_i^*|\mathbf{u}_i, r) = \min_{\mathbf{b}} \mathcal{L}(f_\theta(\mathbf{u}_i, \mathbf{b}), r) = \min_{\mathbf{b}}(\frac{1}{2}(r - U_i^\top(\sum_{j=1}^{|V|} w_{j,i}^v \mathbf{V}_j))^2). \quad (10)$$

The optimized $\mathbf{b}_i^* = (w_{1,i}^*, w_{2,i}^*, ..., w_{|V|,i}^*)$ is a weight vector that is distilled from the pretrained item embedding $\mathbf{V}$. We name it the "the rating preference of user $u_i$" since each element $w_{j,i}^*$ is a distilled weight from item embedding $\mathbf{V}$ that is optimized under the condition of fixed $\mathbf{u}_i$. It may raise concerns that most users only rate a few items. Why do we directly inverse the weight vectors $\mathbf{b}$ about all items? Intuitively, while an individual user might overlook numerous items, many other users have rated these neglected items. As a result, these ratings contribute to the creation of well-generalized item embeddings. The overall item embedding embodies enriched knowledge, thereby facilitating the creation of informative user characteristics.

Suppose we obtain user $u_i$'s rating preference $\mathbf{b}_i^{*,s}$ from source domain, directly transferring the distilled $\mathbf{b}_i^{*,s}$ to the target domain is deemed inappropriate due to the absence of item intersection and the distinct item embedding sizes between the two domains. To render the distilled $\mathbf{b}_i^{*,s}$ transferable, we compress the entire item embeddings by performing a weighted sum on the distilled $\mathbf{b}_i^{*,s}$. Our inspiration is drawn from the attention mechanism [31, 38], which enables each component to contribute distinctively when compressing various elements into a singular representation. In our approach, the distinctive contribution is assigned by the distilled $\mathbf{b}_i^{*,s} = (w_{1,i}^{*,s}, w_{2,i}^{*,s}, ..., w_{|V|,i}^{*,s})$ as follows:

$$\mathcal{P}_i^s = \sum_{j=1}^{|V|} w_{j,i}^{*,s} \mathbf{V}_j. \quad (11)$$

Here, $\mathcal{P}_i^s$ represents the prototype of user $u_i$ in source domain, which is prepared for transfer to the target domain. In this work, the only transferred knowledge from the source domain is the prototype, which is a single representation compressed from the weighted sum of the item embedding matrix $\mathbb{V}$. *This approach avoids using any user sensitive information, thereby mitigating the potential risk of compromising user privacy during knowledge transfer across domains.* The backpropagation mechanism with conditional model inversion on pretrained MF can be viewed on Appendix A.

## 4.3 User Prototype Mapping

The CDR scenario has a necessary knowledge transfer process. Previous methods [34, 41] commonly use overlapping user embeddings to establish a mapping function that facilitates domain alignment. In our study, we aim to mitigate the risk of user-sensitive data leakage by exclusively utilizing prototypes derived from overlapping

users in both domains. To effectively address domain shifts, we implement a prototype mapping function. This learning procedure is formalized as a supervised regression problem, where we aim to minimize the following mapping loss:

$$\min_{\phi} \sum_{u_i \in U^o} \mathcal{L}_{map}(f_{map}(\mathcal{P}_i^s; \phi), \mathcal{P}_i^t). \quad (12)$$

Where $\mathcal{P}_i^s$ and $\mathcal{P}_i^t$ denote the users' prototypes distilled from source and target domains. $\phi$ is the parameter of the mapping function, and the mapping function can be defined as any structure. For simplicity, we adopt a linear layer $f_{map}(\cdot)$ as in [23, 41]. $\mathcal{L}_{map}$ is the mean square error loss.

## 4.4 Search top-k Similar users for Prediction

In existing methods [23, 41], transferred user embeddings from the source domain are often utilized to predict the rating preferences of cold-start users in target domain. In our work, the transfered prototype $\hat{\mathcal{P}}_i^t = f_{map}(\mathcal{P}_i^s; \theta)$, is an item side knowledge distilled from the item embedding matrix. It cannot be directly applied for prediction purposes. Fortunately, as demonstrated by Equation 10, the optimized prototype for user $u_i$ has a potential relationship with its user embedding, i.e., fulfilling the minimal error on $(r - (\mathbf{U}_i^s)^\top \mathcal{P}_i^s)$. Hence, for cold-start user $u_i$ in the target domain, we conduct a search for the top-k users who exhibit strong alignment with the transferred prototype $\hat{\mathcal{P}}_i^t$,

$$\mathbf{M} = top\_k(\min_{\mathbf{U}_i^t}(|r - (\mathbf{U}_i^t)^\top \hat{\mathcal{P}}_i^t|). \quad (13)$$

Where $\mathbf{M} \in \mathbb{R}^{d \times K}$ is the matrix where each column is a searched user embedding in target domain. $K$ is a hyperparameter that denotes the number of collected users. We denote the final prediction of the cold-start user $u_i$ on item $v_j$ by averaging the ratings that are predicted by those top-k users, i.e.,

$$\hat{r} = \frac{1}{K} \sum_{k=1}^{K} \mathbf{M}_k^\top \mathbf{V}_j^t. \quad (14)$$

$\hat{r}$ is the predicted rating. Algorithm is summarized in Appendix B.

## 4.5 Privacy Analysis

In this section, we take a deep analysis on PFCDR framework about the potential privacy leakage problem.

In our proposed PFCDR, user-sensitive data (e.g., user embedding and behaviour logs) never leave the source domain. The only transferred knowledge is the prototype, which is the linear combination of the column vectors of item embedding $\mathbf{V}$ using the elements of $\mathbf{b}_i^*$ as the coefficients. According to Equation 10, when target rating $r$ is known for attackers, user $u_i$'s prototype and its embedding fulfil $\mathbf{U}_i^\top \mathcal{P}_i \approx r$. Therefore, the task of inferring the original user embedding $\mathbf{U}_i$ can be understood as solving the problem of one line linear equation with multiple variables, where the number of variables is the dimension size $d$ of embeddings.

THEOREM 4.1 (PRIVACY-PRESERVING USING PROTOTYPE). *Consider the attacker knows the target rating $r$. Further, suppose the embedding size $d$ of source domain satisfies $d \geq 2$, and the number of non-zero elements within the prototype is larger than or equal to 2. Then, the attacker is unable to infer the user embedding $\mathbf{U}_i$.*

Table 1: Cold-start results over 3 cross-domain tasks. We report the mean results over ten runs. Boldface denotes the best result, and the underline is secondary. ∗ indicates significant 0.05 levels, paired t-test of PFCDR vs. the best baselines.

| CDR Tasks | β | Metric | SDR | | CDR with transferring of user-sensitive data | | | | Privacy-preserving CDR | | | | |
|---|---|---|---|---|---|---|---|---|---|---|---|---|---|
| | | | TGT | LightGCN | CMF | EMCDR | SSCDR | PTUPCDR | FedCDR | P2FCDR | PriCDR-S | PPGenCDR | **PFCDR** |
| Book → Movie | 20% | MAE | 4.1831 | 1.4845 | 1.3632 | 1.1651 | 1.2390 | 0.9970 | 1.3265 | 1.2847 | 1.2947 | 1.2735 | **0.9579** |
| | | RMSE | 4.7536 | 2.0537 | 1.7918 | 1.4548 | 1.6526 | 1.3317 | 1.5922 | 1.5538 | 1.5541 | 1.5293 | **1.2437*** |
| | 50% | MAE | 4.2288 | 1.7643 | 1.5813 | 1.1798 | 1.2137 | 1.0894 | 1.4783 | 1.3467 | 1.3216 | 1.3103 | **0.9742*** |
| | | RMSE | 4.7920 | 2.2216 | 2.0886 | 1.4933 | 1.5602 | 1.4395 | 1.6742 | 1.4812 | 1.5893 | 1.5547 | **1.3113*** |
| | 80% | MAE | 4.2123 | 2.3512 | 2.1577 | 1.3248 | 1.3172 | 1.1999 | 1.5151 | 1.3705 | 1.4413 | 1.3865 | **1.0689*** |
| | | RMSE | 4.8149 | 2.8149 | 2.6777 | 1.6737 | 1.7024 | 1.5916 | 1.7963 | 1.6996 | 1.7317 | 1.6420 | **1.4779*** |
| Book → Music | 20% | MAE | 4.4873 | 1.9585 | 1.8284 | 1.3524 | 1.5414 | 1.2286 | 1.7328 | 1.7281 | 1.6835 | 1.6611 | **1.0717*** |
| | | RMSE | 5.1672 | 2.5714 | 2.3829 | 1.6737 | 1.9283 | 1.6085 | 2.0632 | 1.9899 | 1.9503 | 1.8457 | **1.4548*** |
| | 50% | MAE | 4.5073 | 2.3725 | 2.1282 | 1.4723 | 1.4739 | 1.3764 | 2.4577 | 2.2653 | 2.3169 | 2.1832 | **1.2386*** |
| | | RMSE | 5.1727 | 2.9012 | 2.7275 | 1.8000 | 1.8441 | 1.7447 | 2.6190 | 2.4891 | 2.5834 | 2.4794 | **1.7167** |
| | 80% | MAE | 4.5204 | 3.1243 | 3.0130 | 1.7191 | 1.6414 | 1.5784 | 2.5833 | 2.4432 | 2.5178 | 2.3594 | **1.4947*** |
| | | RMSE | 5.2308 | 3.8677 | 3.6948 | 2.1119 | 2.1403 | **2.0510** | 3.0674 | 2.8115 | 2.9376 | 2.7783 | 2.1358 |
| Music → Movie | 20% | MAE | 4.1077 | 1.5233 | 1.3856 | 1.1669 | 1.2143 | 0.9881 | 1.3290 | 1.1937 | 1.2510 | 1.1958 | **0.9865** |
| | | RMSE | 4.7057 | 1.8104 | 1.7753 | 1.4910 | 1.5231 | 1.3046 | 1.6807 | 1.4648 | 1.5715 | 1.4746 | **1.2658** |
| | 50% | MAE | 4.1232 | 1.8793 | 1.6265 | 1.2114 | 1.2398 | **1.0352** | 1.354 | 1.2267 | 1.2937 | 1.2602 | 1.1014 |
| | | RMSE | 4.6848 | 2.2358 | 2.0613 | 1.5332 | 1.5617 | 1.3566 | 1.7268 | 1.5814 | 1.6271 | 1.5531 | **1.3262** |
| | 80% | MAE | 4.1381 | 2.8828 | 2.6971 | 1.2373 | 1.2564 | 1.1223 | 1.4005 | 1.2763 | 1.3418 | 1.3274 | **1.0712*** |
| | | RMSE | 4.7436 | 3.3547 | 3.1641 | 1.6013 | 1.7176 | 1.5251 | 1.7685 | 1.5966 | 1.7044 | 1.6921 | **1.4285*** |

The proof can be found in Appendix C.

## 5 EXPERIMENTS

We conduct multiple experiments to evaluate the performance of PFCDR and to answer the following research questions: **RQ1** How well does PFCDR perform compared to the state-of-the-art bridge-based CDR approaches in cold-start scenarios? **RQ2** How well does PFCDR perform in more practical scenarios of real-world recommendations? **RQ3** What are the effects of different hyperparameters, and why might PFCDR perform better?

### 5.1 Experimental Setup

**Datasets.** Following the most existing works [16, 37, 41], we use the real-world public dataset for experiments, namely the Amazon review dataset [1]. Specifically, we adopt the Amazon 5-cores dataset, where each user or item has at least five ratings. Following [16, 41], we choose the three commonly used categories: movies_and_tv (Movie), cds_and_vinyl (Music), and books (Book). Then, we define three CDR scenarios: Task 1: Book → Movie, Task 2: Book → Music, and Task 3: Music → Movie. The detailed statistics of datasets are shown in Appendix D.

**Task Settings.** Following [23, 41], to evaluate the effectiveness of PFCDR on cold-start scenario, we randomly remove all ratings of partially overlapping users in the target domain as the test set and the other overlapping users are used for training the prototype bridge function. We assign the test (cold-start) users proportions $β$ as 20%, 50%, and 80% of total overlapping users, respectively.

**Evaluation Metrics.** For easy comparison, we follow [23, 41] to adopt Mean Absolute Error (MAE) and Root Mean Square Error (RMSE) as the metrics.

**Baselines.** Note that PFCDR is a bridge-based method that transfers insensitive information for CDR tasks to protect sensitive user information. To emphasize that we not only meet the privacy requirements but also achieving significant performance

¹http://jmcauley.ucsd.edu/data/amazon/

gain. We compare PFCDR with the following three categories of methods: Single-Domain Recommendation, Cross-Domain Recommendation with user-sensitive information from the source domain, and Privacy-Preserving Cross-Domain Recommendation.

*Single-Domain Recommendation(SDR):* TGT denotes the target MF model, which is training with only target domain data. Light-GCN [12] is a graph convolution method, which ignores the nonlinear activation function and the convolution filter parameter matrix to capture the collaborative signal.

*CDR with user-sensitive information:* CMF [27] is an extension of Matrix Factorization (MF), which shares the latent factors of entities across the source and target domains. EMCDR [23] is a popular embedding-and-mapping framework for handling CDR. SSCDR [16] is a CDR framework for cold-start problems based on a semi-supervised approach, which utilizes both source and target data for training metric space. PTUPCDR [41] adopts the sequential interaction items in the source domain to train a meta network to achieve personalized transfer of user preferences.

*Privacy-Preserving CDR:* FedCDR [24] is a privacy-preserving federated CDR model designed for individual customer scenarios. It builds a cross-domain embedding transformation model on the server side. P2FCDR [5] is a privacy-preserving CDR framework that enhances information fusion at the feature level. PriCDR-S [4] is a two-stage based privacy-preserving CDR framework to protect user-sensitive data leakage. PPGenCDR [18] devotes to stably modeling the distribution of private data in source domain by stable privacy-preserving generator module.

**Parameter Settings.** We implement all algorithms in PyTorch and train on a single NVIDIA Quadro RTX5000 (16GB memory). The parameter settings of PFCDR are of four stages: 1) **Pretraining stage** is to learn latent spaces for source and target domain. In this stage, the learning rate for the Adam optimizer is tuned by grid search within {0.001, 0.005, 0.01, 0.02, 0.1}. The dimension of embedding is 10. For fair comparisons, baselines and PFCDR share the parameter of source and target models. 2) **Inversion stage** is

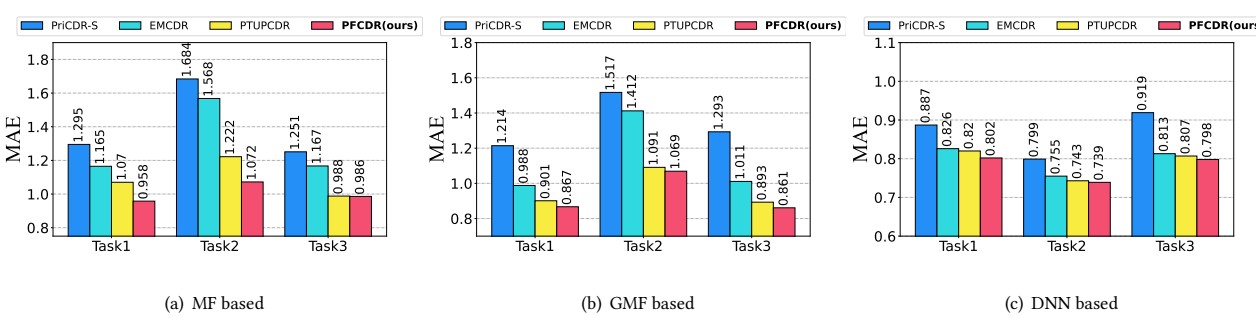

Figure 2: Applying PriCDR-S, EMCDR, PTUPCDR, and PFCDR upon three base models (a) MF, (b) GMF and (c) YouTube DNN.

to distil the prototypes from the source and target domains. In this stage, we optimize the random initialized $\mathbf{v}^*$ with Binary Cross Entropy (BCE) loss and Adam [5] optimizer with learning rate $1e$-4 and weight decay $1e$-4. The target label is 5 during the optimizing process. 3) **Prototype mapping stage** trains bridge functions via the prototypes from both domains. Note that the architecture of the bridge function for prototypes is a one-layer MLP which is the same as the other bridge based-methods (EMCDR, DCDCSR, and SSCDR) for a fair comparison. Then, we employ the same fully connected layer for the remaining bridge-based methods. 4) **Top-k matching stage** is to search the well-matched $k$ users in target domain for further prediction propose. The only hyperparameter is $K$ which denotes the number of selected users for prediction. We find it is an important parameter that needs to be tuned, and details about $K$ has discussed in Section 5.4.1.

## 5.2 Performance Comparison (RQ1)

We conduct a quantitative experiment of PFCDR's cross-domain performance, focusing on three CDR tasks within cold-start scenarios. This evaluation involves comparing PFCDR against three established categories of methods: SDR [12], CDR incorporating user-sensitive information [16, 23, 27, 39, 41], and privacy-preserving CDR [4, 5, 24]. Table 1 shows the results with different $\beta$ values, which are the test fractions of the overlapping users. The best performance is shown in boldface, and the underline is denoted as secondary. ∗ is the 0.05 level paired t-test of PFCDR vs. the best baseline. According to the experimental results, we have the following observations: (1) The performance of TGT is unsatisfying since it is a MF-based single-domain model which only uses data from the target source. Compared with the TGT and LightGCN methods, although LightGCN adopts a more sophisticated architecture, all other MF-based cross-domain methods can adopt an auxiliary domain to mitigate the data sparsity problem. (2) CMF directly fuses the data of the two domains for training, while CDR methods design different bridge functions to bridge domains. It is shown that CDR methods outperform CMF in most cases. The reason lies in the fact that the CMF disregards the possibility of domain shift by simply merging data from different domains into one domain, whereas CDR methods utilize a bridge function to convert the source user embedding to the target feature space. (3) Although PFCDR transfers prototypes from source domain to target domain without any user-sensitive data, it outperforms the best baseline PTUPCDR, which utilizes both user-item interactions and trained user embeddings for transferring. In contrast to traditional methods

such as PTUPCDR or privacy-preserving approaches like PriCDR and P2FCDR, which heavily depend on user embeddings or user rating information for knowledge transfer, PFCDR maximizes the utilization of model knowledge contained within item embeddings to distill transferable information.

## 5.3 Generalization Experiments (RQ2)

The majority of bridge-based CDR literature [23, 39] mainly focuses on the bridge function and primarily tests it on MF. Hence, to testify the generality of our proposed PFCDR and other baselines, we implement PriCDR-S, PFCDR, PTUPCDR, and EMCDR upon two more complicated neural models. In this paper, We adopt GMF [13] and YouTube DNN [6] to replace MF to train source and target domain. GMF is a generalized version of MF since it assigns various weights to different dimensions for the dot-product prediction function. YouTobe DNN is a two-tower neural model. In order to more effectively compare the user embedding in various models, both GMF and YouTobe DNN directly adopt the user embedding to train the bridge functions. We conduct evaluations on both non-neural (MF) and neural models (GMF and YouTobe DNN) with $\beta$ = 20%, while other experimental settings are consistent with Section 5.1. As shown in Figure 2, we have the following observations: (1) All bridge-based methods can be smoothly implemented on different models, while the PriCDR-S, EMCDR and PTUPCDR effectively improve the performance in GMF and YouTube DNN. At the same time, the GMF and YouTube DNN are two advanced neural models for sophisticated industrial recommendations, and they outperform the MF. (2) As we can see, the PFCDR still achieve the best performance among different CDR methods. Specifically, without relying on source data information, the PFCDR approach is a more suitable choice for real-world commercial corporations.

## 5.4 Hyperparameters and Visualization (RQ3)

In this section, we conduct extensive experiments to analyze the impact of various hyperparameters and to explore the performance improvements from a visualization perspective.

*5.4.1 Parameter K.* Parameter $K$ is a hyperparameter that needs to be tuned in the Top-k matching stage. We analyze how it affects the final performance on different models (MF, GMF, and YouTube DNN) with different tasks (as shown in Figure 3(a)-(c)). The x-axis denotes the selected user numbers for predicting, and the y-axis denotes the MAE on different tasks. According to the experimental results, we have the following observations: (1) For task 1 (book → Movie) and task 3 (Music → Movie) the more selected user for

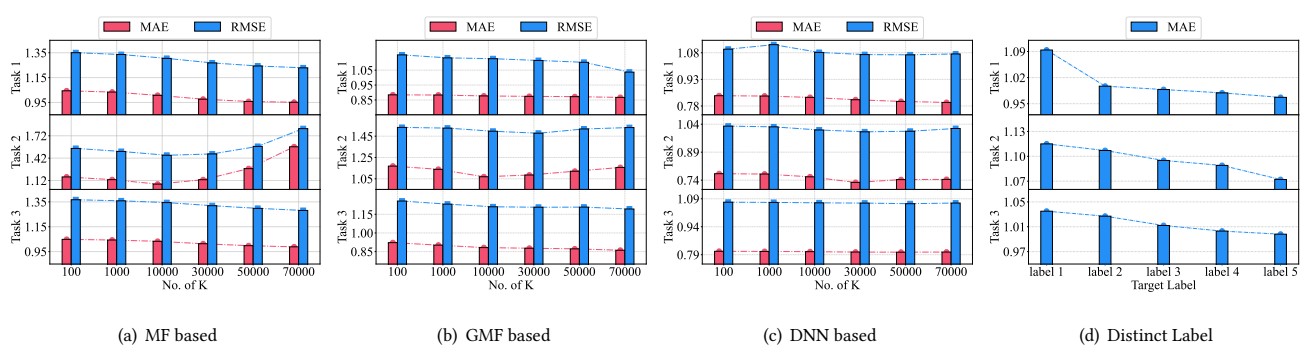



(a) MF based     (b) GMF based     (c) DNN based     (d) Distinct Label

**Figure 3: (a)-(c) is ablation study on parameter $K$, and (d) is ablation study on distinct target label.**

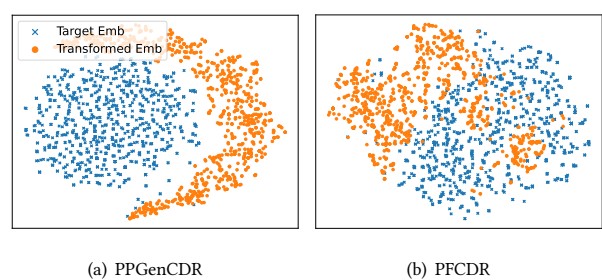

(a) PPGenCDR      (b) PFCDR

**Figure 4: t-SNE visualization of randomly sampled user embeddings from target-domain feature space and transformed user embeddings.**

predicting the better performance while the task2 (Book → Music) reaches its best performance between 10k and 30k users. This is because the domain includes domain-shared and domain-specific knowledge [3]. The source and target domains in task1 and task3 have more domain-shared knowledge. For task 1 (book → Movie), we know that many movies are remade from books, and some books are inspired by movies. For task 3 (Music → Movie), we know that plenty of classical music is the episode in movies. (2) The impact of $K$ on final performance is more obvious on MF-based PFCDR, but with a more sophisticated base model such as GMF and DNN, the impact is reduced.

*5.4.2 Robustness under Variant Target Ratings in Inversion Stage.* When we distil the user prototype with conditional model inversion, the selection of target ratings is a tuneable parameter. Hence, to verify how the target rating impacts the generated prototypes and finally change the prediction results of CDR (as shown in Figure 3(d)). We generate users' prototypes under variant target labels from 1-5, while the other experimental settings are the same as in Section 5.1. From the observation of Figure 3(d), we have the following insightful observation: (1) With different target labels as conditions, the inversion stage can provide well and similar performance in the final CDR, which shows the robustness of our proposed PFCDR. (2) With the target rating enlargement(from dislike to like), the overall performance is gradually improved. The main reason would be that a more positive rating can reflect a more precise user's prototype about what he/she is most interested in.

*5.4.3 Latent Factor Visualization.* We analyze the embeddings on the target domain feature space to further investigate why PFGenCDR outperforms the PPGenCDR. But in our PFCDR, we do not transfer the user embeddings to the target domain. To further compare in a visual latent space, we average sum the Top-k selected user embeddings into one representation. We employ t-distributed Stochastic Neighbor Embedding (t-SNE) in Scikit-learn to visualize the transferred user embedding learned by PFCDR and PPGenCDR on Task 1 with $\beta = 20\%$. Figure 4 (a) and (b) denote the embedding visualization of EMCDR and PFCDR. The blue points are the target embeddings learned with both training and test users and are regarded as the ground truth, while the orange point is the transferred user embeddings mapping from PPGenCDR and PFCDR, respectively. We random sample 1000 points of the test users to illustrate the visualization.

In general, the closer the distance between transferred embedding and target embedding (ground truths), the better. From the Figure 4, we have the following observations: (1) The transformed embeddings from PFCDR are closer to the target embeddings, indicating that the average of the selected top-k user embeddings in PFCDR aligns well with the target embeddings. (2) The transformed embeddings from PFCDR are more scattered across the target domain than PPGenCDR. This difference is primarily because GAN-based methods often generate a limited subset of possible outputs, failing to capture the diversity of the target distribution.

## 6 CONCLUSION

This paper presents a novel PFCDR system to realize cross-domain recommendation without transferring any user-sensitive data in the source domain. To the best of our knowledge, we are the first to apply the data-free knowledge distillation in CDR tasks. We propose a new data format and a new learning paradigm to overcome the inconsistent gradient optimization process. Simultaneously, the conditional model inversion is proposed to distil the user's prototype. Then, we train a bridge function to alleviate the domain shift problem across domains. Finally, a search mechanism has been leveraged to select top-k well-matched users in the target domain, and we use their average behaviors to predict a new cold-start user. We conducted extensive experiments on real-world datasets to evaluate the proposed PFCDR, and the results successfully demonstrate the effectiveness of PFCDR in dealing with the cold-start problem.

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

 

# A  BACKPROPAGATION ON NEW DATA FORMAT

Suppose we learn $\mathbf{b}^*$ with Stochastic Gradient Descent (SGD) [2], under the new learning paradigm introduced in Equation 9, each element of $\mathbf{b}^*$ can be updated using the following formula:

$$
\begin{aligned}
(w_l)_{t+1} &= (w_l)_t - \eta \nabla \mathcal{L}(f_\theta(u_i, b), r) \\
&= (w_l)_t - \eta \nabla (\frac{1}{2}(r - \mathbf{U}_i^\top (\sum_{j=1}^{|V|} \mathbf{V}_j w_j^v))^2) \\
&= (w_l)_t - \eta [(r - \mathbf{U}_i^\top (\sum_{j=1}^{|V|} \mathbf{V}_j w_j^v))(-\mathbf{U}_i^\top \mathbf{V}_l)] \\
&= (w_l)_t - \eta [(r - \mathbf{U}_i^\top (\sum_{j=1,j\neq l}^{|V|} \mathbf{V}_j w_j^v + \mathbf{V}_l w_l^v))(-\mathbf{U}_i^\top \mathbf{V}_l)],
\end{aligned}
\tag{15}
$$

where $t$ is current epoch and $\eta$ is the learning rate. The $\mathbf{U}_i^\top \mathbf{V}_l$ is a constant. The update of $w_l^v$ in one epoch is dominated by the predefined target rating, user $u_i$'s embedding vector, and pretrained item embedding matrix in the source model. Thus, when we utilize different user $u_i$'s embedding vectors as constraints, the weight vector $\mathbf{b}$ is changed to match the embedding vector $\mathbf{U}_i$ by monotonically decreasing the loss.

# B  ALGORITHM

As illustrated in Algorithm.1, the overall training Algorithm of PFCDR can be divided into four steps: the inversion stage, prototype mapping stage, top-k matching and prediction stage. After training, our method can significantly alleviate the cold-start recommendation problem in the target domain without leakage of any user-sensitive information from the source domain.

**Inversion stage:** Most existing CDR methods [4, 24] highly depend on transferring user-sensitive knowledge (user behaviors log or pretrained user embedding) from the source domain. Therefore, their main contributions are adopting some privacy-preserving strategies to minimize the risk of leaking user-sensitive information. However, in our proposed framework, user-sensitive information is never been used. Since we adopt conditional model inversion on both the source and target domains to distil the rating preference $\mathbf{v}^*$ from the item embedding (line 1-7), and then compress the item embedding matrix by performing weighted sum on $\mathbf{v}^*$ to generate and save the users' prototypes (line 8-9). In this stage, we adopt mini-batch training from the memory-economic perspective.

**Prototype mapping stage:** After obtaining the users' prototypes $\mathcal{P}_o^s$ and $\mathcal{P}_o^t$ from source and target domain, we adopt the a linear layer mapping function to bridge the domain shift across domains (line 15).

**Top-k matching stage:** When an extreme cold-start user $u_i$ comes into the target domain (in our CDR assumption, the new user has interactions in the source domain), we transfer the user prototype $\mathcal{P}_i^s$ from source domain to target domain $\hat{\mathcal{P}}_i^t$ via the mapping function (line 18) and search the top-k well-matched users in target domain (line 19).

**Prediction stage:** We predict the interactions of cold-start user $u_i$ in target domain by the averaging rating of those collected topk users in target domain (line 20).

---

**Algorithm 1:** PFCDR

**input** : Pretrained source domain $f_{\theta^s}$, pretrained target domain $f_{\theta^t}$; overlapped user set $U^o$; the threshold $\epsilon$ for stopping the optimization of $\mathbf{v}^*$; randomly initialize a batch of target rating $r$; $L$ is the training epochs for mapping function;

**output** : The prediction of item $\mathbf{V}_j^t$;

/* Inversion stage */
1  Randomly initialize $|U^o|$ number of $\mathbf{v}$ with Eq.(5);
2  **while** *True* **do**
3  $\quad$ $\hat{r} \leftarrow f_{\theta^s}(\mathbf{u_i}, \mathbf{v_n})$;
4  $\quad$ $loss_n \leftarrow \mathcal{L}(\hat{r}, r)$;
5  $\quad$ **if** $loss_n \geq \epsilon$ **then**
6  $\quad\quad$ $\mathbf{v}_{n+1} = \mathbf{v}_n - \eta \frac{\alpha loss_n}{\alpha \mathbf{v}_n}$;
7  $\quad$ **else**
8  $\quad\quad$ $\mathbf{v}^* \leftarrow \mathbf{v}_n$;
9  $\quad\quad$ save prototype $\mathcal{P}_i^s$ with Eq.(11);
10 $\quad\quad$ **break**;
11 $\quad$ **end**
12 **end**
13 Same operation (from line 1-11) on target domain to generate prototype $\mathcal{P}_i^t$;
/* Protopype mapping stage */
14 **for** $epoch \leftarrow 0$ **to** $L$ **do**
15 $\quad$ Training prototype mapping function with Eq.(12);
16 **end**
/* Top-k matching stage */
17 A cold-start user $u_i$ ($u_i \in U^s$ & $u_i \notin U^t$) comes into target domain;
18 $\hat{\mathcal{P}}_i^t \leftarrow f_{map}(\mathcal{P}_i^s)$;
19 Select top-k users with Eq.(13);
/* Prediction stage */
20 predicted rating $\hat{r}$ of item $\mathbf{V}_j^t$ with Eq.(14);

---

# C  PRIVACY ANALYSIS

*Theorem 4.1 (PRIVACY-PRESERVING USING PROTOTYPE). Consider the attacker knows the target rating $r$. Further, suppose the embedding size $d$ of source domain satisfies $d \geq 2$, and the number of non-zero elements within the prototype is larger than or equal to 2. Then, the attacker is unable to infer the user embedding $\mathbf{U}_i$.*

PROOF. For ease of exposition, we prove the theorem in two cases:1) $d = 2$; and 2) $d > 2$.

**Case 1.** Given the embedding size $d = 2$ of source domain, we have $\mathbf{U}_i^\top \mathcal{P}_i = w_1 \mathcal{P}_{i,1} + w_2 \mathcal{P}_{i,2} = r$. Hence, the analytic expression of $w_1$ and $w_2$ are $w_1 = \frac{r - w_2 \mathcal{P}_{i,2}}{\mathcal{P}_{i,1}}$ and $w_2 = \frac{r - w_1 \mathcal{P}_{i,1}}{\mathcal{P}_{i,2}}$. So value of one variable $w_1/w_2$ decides the value of the other variable $w_2/w_1$. We can put infinitely many values for one variable which will result in infinitely many values for the second variable.

**Case 2.** Given the embedding size $d = N$ of source domain, where $N > 2$, we have $\mathbf{U}_i^\top \mathcal{P}_i = w_1 \mathcal{P}_{i,1} + w_2 \mathcal{P}_{i,2} + ... + w_N \mathcal{P}_{i,N} = r$. Hence, the analytic expression of $w_n$ is $w_n = \frac{r - (\sum_{1 \leq m \leq N and m \neq n} w_m \mathcal{P}_{i,m})}{\mathcal{P}_{i,n}}$. We

**Table 2: The statistic of cross-domain tasks, overlap denotes the number of overlapping users.**

| CDR Tasks | Domain | | Item | | User | | | Rating | |
|---|---|---|---|---|---|---|---|---|---|
| | Source | Target | Source | Target | Overlap | Source | Target | Source | Target |
| **Book → Movie** | Book | Movie | 367,982 | 50,052 | 37,388 | 603,668 | 123,960 | 8,898,041 | 1,697,533 |
| **Book → Music** | Book | Music | 367,982 | 64,443 | 16,738 | 603,668 | 75,258 | 8,898,041 | 1,097,592 |
| **Music → Movie** | Music | Movie | 64,443 | 50,052 | 18,031 | 75,258 | 123,960 | 1,097,592 | 1,697,533 |

**Table 3: Computation complexity and storage usage.**

| Metrics | EMCDR | PriCDR | PF2CDR | FedCDR | **PFCDR** |
|---|---|---|---|---|---|
| Training Time (Min.) | **3** | 16 | 35 | 43 | 19 |
| Testing Time (Sec.) | **37** | 1749 | 45 | 59 | 149 |
| Storage Usage (Mega Byte) | 5 | 768 | 243 | 419 | **5** |

can put infinitely many values for variable $w_n$ which will result in infinitely many values for the other variables $\{w_1, w_2, ..., w_N\}\backslash\{w_n\}$.

Therefore, Theorem 4.1 holds.                    □

Hence, the risk of leakage the user embedding is kept to a minimum in source domain when the dimension of user embedding is equal or greater than 2.

## D    DATASET DESCRIPTION

We adopt the Amazon 5-cores dataset, where each user or item has at least five ratings. Following [16, 41], we choose the three commonly used categories: movies_and_tv (Movie), cds_and_vinyl (Music), and books (Book). Then, we define three CDR scenarios: Task 1: Book → Movie, Task 2: Book → Music, and Task 3: Music → Movie. We list details in Table 2.

Unlike existing works that typically assume the data size in the source domain to be significantly larger than that in the target domain, we also consider scenarios where the target domain has fewer ratings compared to the source domain. While many prior studies only evaluate a subset of the dataset, we utilize the entire dataset to simulate a more practical scenario.

## E    COMPUTATION COMPLEXITY AND STORAGE USAGE ANALYSIS

To systematically analyze the computation complexity and storage consumption of the PFCDR method, we compare it with existing baseline methods on the music →movie task, where $\beta$ = 20%. To compare the temporal and spatial advantages of PFCDR with the baseline methods throughout the entire lifecycle, we select training time, testing time, and storage usage as the comparison metrics. The conclusions based on Table 3 are as follows:

(1) In terms of training time, the EMCDR method, which directly transfers user embeddings from the source domain to the target domain without employing any privacy protection techniques and only requires training a one-layer MLP as the mapping function, has the shortest training time. In contrast, the PriCDR method introduces additional time consumption by converting the source domain's rating matrix into a noisy matrix using differential privacy techniques. Both the PF2CDR and FedCDR methods, based on the federated learning framework, require joint training of the source and target domains, leading to a longer convergence process and the

longest training times. Our PFCDR only needs to extract prototype knowledge from the pretrained model, resulting in a training time that is shorter than PF2CDR and FedCDR but longer than EMCDR and PriCDR.

(2) In terms of testing time, the PriCDR method takes the longest. This is because the target domain needs to use the noisy matrix obtained during the training phase to train the target model. Since the noisy rating matrix is in floating-point format, it significantly increases the training time of the target model. The EMCDR method has the shortest testing time because it directly uses the user embeddings transferred from the source domain for testing in the target domain. The PFCDR method requires matching top-k users in the target domain. Although this matching process extends the testing time, each cold-start user only needs to be matched once, after which the recorded top-k users can be used for predictions.

(3) In terms of storage requirements, the PFCDR and EMCDR methods have the lowest storage demands. PFCDR only needs to store prototype vectors that have identical dimension of the user embeddings and additionally trains a one-layer MLP model as the prototype mapping fuction. The PriCDR method has the highest storage usage due to the need to store the noisy floating-point rating matrix. Both PF2CDR and FedCDR methods require retraining the model, so their main storage consumption comes from the model parameters. The FedCDR method, which uses a graph neural network, also needs to store the graph's node and edge information. Due to the more complex network structure, its storage usage is slightly higher than that of PF2CDR.

