# OpenReview forum: "Privacy-Friendly Cross-Domain Recommendation via Distilling User-irrelevant Information"
_ACM.org/TheWebConf/2025/Conference — WWW 2025 Oral_

### Official Review · Reviewer_qHT3 · 2024-11-29

**Novelty:** 5
**Technical Quality:** 5

**Review:**

This paper proposes Privacy-Friendly Cross-Domain Recommendation (PFCDR), which tackles the cold-start problem without transferring sensitive user data, which often requires distortion for privacy protection. Instead, PFCDR distills transferable knowledge from insensitive item embeddings, termed prototypes, using a conditional model inversion mechanism. This novel approach creates a new data format and learning paradigm to extract prototypes from traditional recommendation models, bridging domain gaps while preserving privacy. Extensive experiments on large datasets demonstrate its effectiveness, and the code is publicly available.

Pros:

1. Privacy protection is important for Cross-Domain Recommendation.
2. The experimental results look promising.
3. This paper is easy to understand.

Cons:

1. The proposed method cannot provide a formal privacy protection guarantee like DP.
2. Only one, toy, small dataset is used in experiments.
3. Only some out-of-dated RS methods like MF and GCN are tested. More new and SOTA RS methods should be included in experiments.

**Questions:**

1. Is it possible that some privacy guarantee can be provided for the proposed methods?

2. Why not try some real-world and large-scale datasets? Amazon dataset is for review rating prediction, whose scenario is very different from the product RS systems.

3. Can you try some SOTA recommendation methods like those based on LLMs?

**Reviewer Confidence:**

2: The reviewer is willing to defend the evaluation, but it is likely that the reviewer did not understand parts of the paper

**Scope:**

3: The work is somewhat relevant to the Web and to the track, and is of narrow interest to a sub-community

---

### Official Review · Reviewer_ezTX · 2024-12-02

**Novelty:** 5
**Technical Quality:** 5

**Review:**

To address the problem of privacy-preserving cross-domain recommendation (CDR), this paper introduces a method called PFCDR. PFCDR uses a technique of distilling insensitive item embeddings as prototypes to address challenges in knowledge transfer while preserving user privacy. Experimental results on multiple datasets demonstrate the effectiveness of the proposed method.

Pros:

1. The motivation of the paper is clearly introduced, emphasizing the trade-offs in privacy-preserving CDR and the focus on using item embeddings rather than sensitive user data.

2. The authors provide source code and datasets, significantly enhancing the reproducibility of the study.

3. The proposed method is well-founded, and the experiments validate its effectiveness in cold-start recommendation scenarios.

Cons:

1. The novelty of the method is uncertain due to the omission of several important references that overlap with the topic of privacy-preserving cross-domain recommendation, such as [1,2,3]. The authors should cite these works and provide a thorough comparison to clarify the unique contributions of their approach.

2. The paper evaluates the method only on rating prediction tasks. However, top-k recommendation, which is more common in practical recommender systems, is not explored. This limitation reduces the practical relevance of the study.

[1] Guo et al., "Prompt-enhanced Federated Content Representation Learning for Cross-domain Recommendation," WWW 2024.

[2] Liu et al., "FedCT: Federated Collaborative Transfer for Recommendation," SIGIR 2021.

[3] Yang et al., "Federated Graph Learning for Cross-Domain Recommendation," ArXiv 2024.

**Questions:**

1. How does the proposed method differ from the works cited above in terms of privacy-preserving mechanisms and cross-domain recommendation techniques? Can the authors provide a detailed comparative analysis?

2. How does the proposed method perform on top-k recommendation tasks?

**Reviewer Confidence:**

4: The reviewer is certain that the evaluation is correct and very familiar with the relevant literature

**Scope:**

4: The work is relevant to the Web and to the track, and is of broad interest to the community

---

### Official Review · Reviewer_MLK8 · 2024-12-02

**Novelty:** 5
**Technical Quality:** 6

**Review:**

This paper introduces PFCDR, a framework designed to preserve user information privacy in cross-domain recommendation. Instead of transferring user embeddings, PFCDR extracts user knowledge, called prototypes, using a conditional model inversion mechanism. These prototypes are used to bridge the source and target domains through a mapping function and address the cold-start problem by identifying top-k users in the target domain that align with the transferred prototypes.

Pros: 1. The paper has a very clear writing structure. 2. The idea of transferring user prototype instead of user embedding is interesting and potentially impactful.

Cons: 1. The distilled user prototype is not user-irrelevant. It has some relations to the user. The title is causing confusion for understanding the content. 2. The figure of the framework is hard to understand and it needs to be reorganized. It shows the details instead of overall structure and idea for each module. 3. The new data format has identical dimension to the one-hot representation, hence it greatly increases the computational cost, since each user prototype is computed based on the whole item embedding matrix. Also, it is better to clearly mention the size of $\mathbf{a}$ and $\mathbf{b}$.

**Questions:**

1. Could you explain the elements in the figures clearly?
2. How is the computation performance for PFCDR? Is this model able to deployed in real cases when there are much more items than users, and new items are created and old items are deleted every day?
3. Some performance of PFCDR is not better than the baseline. Are there any explanations for these results?

**Reviewer Confidence:**

4: The reviewer is certain that the evaluation is correct and very familiar with the relevant literature

**Scope:**

4: The work is relevant to the Web and to the track, and is of broad interest to the community

---

### Official Review · Reviewer_Q2qD · 2024-12-02

**Novelty:** 5
**Technical Quality:** 5

**Review:**

This paper introduces a novel approach to cross-domain recommendation with a focus on privacy-preserving mechanisms. It helps to achieve cross-domain recommendations without transferring any user-sensitive data in the source domain and mitigate the risk of compromising user-sensitive privacy while achieving satisfactory performance.

The Pros of the paper:
1. it introduces an innovative privacy mechanism by focusing on transferring insensitive item embeddings (prototypes) rather than user-sensitive data.
2. It provides extensive experiments across multiple real-world datasets and scenarios to validate the proposed method's effectiveness in terms of both privacy and performance.
3. It provides solid comparisons with a wide range of baselines, highlighting the model's superiority.

The Cons of the paper:
1. The proposed method involves multiple stages (model inversion, prototype mapping, top-k matching), which could be challenging to implement and optimize in practical systems.
2. The performance heavily relies on the quality of item embeddings. Poor embeddings may lead to suboptimal prototype generation and, consequently, degraded recommendation quality.
3. While the method avoids sharing user-sensitive data, the conditional model inversion and prototype mapping stages may introduce computational overhead, especially for large-scale datasets.

Overall, while the paper presents a significant advancement in privacy-preserving CDR, it leaves room for addressing practical challenges, scalability, and broader applicability.

**Questions:**

1. Could you clarify how prototypes differ from traditional item embeddings in terms of their structure and transferability? How does the compression step ensure the preservation of critical information for recommendations?

2. How robust is the method to variations in the target rating used during the model inversion stage? Have you tested scenarios where the chosen rating does not align with user behavior in the target domain?

**Reviewer Confidence:**

3: The reviewer is confident but not certain that the evaluation is correct

**Scope:**

4: The work is relevant to the Web and to the track, and is of broad interest to the community

---

### Official Review · Reviewer_fVdo · 2024-12-02

**Novelty:** 4
**Technical Quality:** 4

**Review:**

This paper proposes a privacy-friendly cross-domain recommendation method, which ensures user privacy while significantly improving recommendation performance by extracting user “prototypes.” The motivation of the paper is clear, and the method is sound, but there are still some minor issues that leave me confused.


Strengths:
1. The article is very well-written, and the structure is well-organized.
2. The experiments are solid and demonstrate significant performance improvements.
3. This paper introduces a novel method for cross-domain recommendation (CDR) that focuses on extracting and transferring user-irrelevant information, specifically item prototypes, rather than sensitive user embeddings or behavior logs. This approach effectively addresses privacy concerns while maintaining the utility of the recommendation system.
4. This paper proposes a conditional model inversion mechanism to accurately distill prototypes for individual users. This mechanism is robust to variations in target ratings, showing consistent performance improvements as the target rating increases from dislike to like. This ensures that the extracted prototypes are reliable and representative of user preferences.


Weakness:
1. The paper states that the prototypes are generated by weighting and combining the embeddings of items the user has interacted with, claiming that this hides the user's private preferences. However, it seems that the set of interacted items itself can strongly reflect user preferences. I am confused as to why using all the items a user has interacted with still does not reveal their private preferences.
2. The paper does not explicitly explain the specific role of Knowledge Distillation in the proposed method. I am curious whether the purpose of Knowledge Distillation is to better hide user-private information embedded in the item embeddings or if it serves some other function.
3. The bridge function is used to match user prototypes across different datasets. I would like to know the accuracy of this function in matching users successfully. This accuracy seems closely related to the final recommendation performance.
4. The conditional model inversion mechanism, while innovative, can be computationally intensive and complex to implement. The process of optimizing the randomly initialized vector b for each user and target rating requires significant computational resources, which might be a barrier for real-world applications with large datasets and limited computational power.
5. The effectiveness of the proposed method is highly dependent on the amount of domain-shared knowledge between the source and target domains. For tasks where the domains have less overlap (e.g., Book → Music), the performance improvement is less pronounced compared to tasks with more shared knowledge (e.g., Book → Movie). This limitation suggests that the method may not be equally effective across all types of cross-domain recommendation scenarios.

**Questions:**

1. My main concerns is, what is the benefit by introducing a*, b* compared to using vanilla matrix factorization to directly learn U and V? In addition, if we use some collabrative filtering based methods such as matrix factorization, lightGCN or neural collabrative filtering (NCF), is the Theorem 4.1 still hold without the projection, i.e., the coefficient a* and b*? Taking NCF as an example, I think we cannot identify the user embedding in the presence of multiple layers of MLP.

2. This paper mainly uses Amazon's datasets for experiments, but the results obtained using only a single website's datasets are not very convincing. Could the author provide some experimental results on datasets from other websites?

3. Could the authors provide some insights or suggest possible feasible solutions based on the proposed method when the amount of domain shared knowledge between the source and target domains is small?

**Reviewer Confidence:**

3: The reviewer is confident but not certain that the evaluation is correct

**Scope:**

3: The work is somewhat relevant to the Web and to the track, and is of narrow interest to a sub-community